# Resilience and Attachment in Patients with Major Depressive Disorder and Bipolar Disorder

**DOI:** 10.3390/jpm13060969

**Published:** 2023-06-08

**Authors:** Ambra Craba, Giuseppe Marano, Georgios D. Kotzalidis, Carla Avallone, Francesco Maria Lisci, Maria Luigia Crosta, Antonino Callea, Laura Monti, Domenico De Berardis, Carlo Lai, Marta Balocchi, Ilenia Sessa, Désirée Harnic, Gabriele Sani, Marianna Mazza

**Affiliations:** 1Department of Geriatrics, Neuroscience and Orthopedics, Fondazione Policlinico Universitario Agostino Gemelli IRCCS, 00168 Rome, Italy; ambra.craba@hotmail.it (A.C.); giuseppemaranogm@gmail.com (G.M.); avallonecarla@yahoo.it (C.A.); fmlisci@gmail.com (F.M.L.); marialuigiacrosta@gmail.com (M.L.C.); marta.balocchi@hotmail.it (M.B.); ileniasessa3@gmail.com (I.S.); gabriele.sani@unicatt.it (G.S.); marianna.mazza@policlinicogemelli.it (M.M.); 2Department of Psychiatry, Università Cattolica del Sacro Cuore, 00168 Rome, Italy; 3Department of Neurosciences, Mental Health, and Sensory Organs (NESMOS), Sapienza University of Rome, 00189 Rome, Italy; 4Department of Human Sciences, LUMSA University, 00193 Rome, Italy; a.callea@lumsa.it; 5UOS Psicologia Clinica, Governo Clinico, Fondazione Policlinico Universitario Agostino Gemelli IRCCS, 00168 Rome, Italy; laura.monti@policlinicogemelli.it; 6Department of Mental Health, ASL 4, 64100 Teramo, Italy; domenico.deberardis@aslteramo.it; 7Department of Dynamic, Clinical Psychology and Health Studies, Faculty of Medicine and Psychology, Sapienza University of Rome, 00185 Rome, Italy; carlo.lai@uniroma1.it; 8Centro Lucio Bini, 00193 Rome, Italy; desiree.harnic@yahoo.com

**Keywords:** major depressive disorder, bipolar disorder, resilience, attachment styles, mood disorders, secure attachment, fearful attachment

## Abstract

Background: Resilience represents one of the fundamental elements of attachment and has often been investigated in mood disorders. This study aims to investigate possible correlations between attachment and resilience in patients with major depressive disorder (MDD) and bipolar disorder (BD). Methods: 106 patients (51 MDD, 55 BD) and 60 healthy controls (HCs) were administered the 21-item Hamilton Depression Rating Scale (HAM-D-21), the Hamilton Anxiety Rating Scale (HAM-A), the Young Mania Rating Scale (YMRS), the Snaith–Hamilton Pleasure Scale (SHAPS), the Barratt Impulsiveness Scale-11 (BIS-11), the Toronto Alexithymia Scale (TAS), the Connor–Davidson Resilience Scale (CD-RISC), and Experiences in Close Relationship (ECR). Results: MDD and BD patients did not significantly differ from each other according to the HAM-D-21, HAM-A, YMRS, SHAPS, and TAS, while they scored higher than HCs on all these scales. Patients in the clinical group scored significantly lower on CD-RISC resilience than HCs (*p* < 0.01). A lower proportion of secure attachment was found among patients with MDD (27.4%) and BD (18.2%) compared to HCs (90%). In both clinical groups, fearful attachment prevailed (39.2% patients with MDD; 60% BD). Conclusions: Our results highlight the central role played by early life experiences and attachment in participants with mood disorders. Our study confirms the data from previous research showing a significant positive correlation between the quality of attachment and the development of resilience capacity, and supports the hypothesis that attachment constitutes a fundamental aspect of resilience capacity.

## 1. Introduction

The term “resilience” was taken from mechanical sciences and introduced into psychology and psychiatry in a similar way to “stress”. In materials science, resilience is related to the ability of a material to revert to its original form after being bent, pressed, or stretched. The founder of the theory of resilience was the clinical psychologist Norman Garmezy [1,2]. Many investigators refined and expanded his theory [3,4]. Resilience is inbuilt in living systems; every person possesses resilience and can strengthen it. It consists of seven columns, which are optimism, acceptance, focus on problem solving, defense mechanisms, forgiveness, responsibility, acquaintance, and planning future. In psychology and psychiatry, resilience is broadly defined as the ability to successfully adapt to adversity, trauma, tragedy, or significant threat [5,6]. It is well known that stress can play an important role in the onset or relapse of bipolar disorder (BD) or major depressive disorder (MDD) [7,8,9]. Individual and psychological responses to the same stressor vary, with differences according to a higher or lower presence of resilience [10]. It has been noted that patients with depressive disorders have lower levels of resilience than healthy individuals [11]. Resilience capacity is also related to severity of depressive symptoms and the response to treatment [12,13]. Studies conducted on patients with BD agree in terms of detecting low levels of resilience and high levels of impulsivity in patients compared to controls [14,15].

Attachment theory postulates that humans have the disposition to build and maintain some intimate social bonds that are critical for good mental health. The term “attachment” refers to the emotional, cognitive, and behavioral processes involved in the formation and preservation of these bonds. According to attachment theory, children develop internal working models of themselves and others on the basis of early interactions and experiences with their caregivers. These models shape their expectations of how their future relational interactions will play out and also define the associated emotional and behavioral responses [16]. Mary Ainsworth experimentally identified three subgroupings of attachment relationships: secure, anxious–avoidant, and anxious–resistant (or ambivalent) attachment styles [17,18]. Insecure attachment styles (anxious/ambivalent or avoidant) seem to be associated with higher levels of psychopathology, including depression, anxiety, and substance use disorders [19]. John Bowlby suggested that the loss of attachment security in early life contributes to the development of depression in adulthood, because the loss of security promotes the formation of negative representations of oneself and the world [20]. Indeed, attachment plays a major part in the development of emotion regulation and models of self, and there are important links between depression, dysfunctional emotion regulation, and dysfunctional schemas about the self [21]. In addition, different clinical studies have highlighted the existence of a positive correlation between resilience and secure attachment, which could be mediated by high levels of self-esteem. Consistently, some studies have confirmed the presence of greater resilience in individuals with secure attachment styles compared to those who have insecure ones [22,23]. A meta-analysis found resilience and secure attachment to be weakly to moderately correlated [24]; however, most studies focused on the outcomes of stress, domestic violence, and other traumatic experiences that may be involved in post-traumatic stress disorder [25,26,27]. One recent study focused only on bipolar disorder and found childhood trauma in more than half of the patient sample; trauma significantly correlated with low resilience and high attachment-related avoidance, as well as attachment-related anxiety [28]. Attachment-related avoidance and anxiety and low resilience were also found to correlate with psychopathology in borderline personality disorder [29], a notoriously trauma-related condition [30,31].

Although the literature shows that attachment is associated inversely with depression and directly with resilience [22,23,32], there are no clinical studies aimed at investigating the relationship between attachment styles and resilience in patients with mood disorders. Starting from the above theoretical considerations, the present observational study proposes the following objectives: to analyze resilience in patients with MDD and BD and non-clinical controls; to analyze attachment styles in patients with MDD and BD and non-clinical controls; and to highlight any correlations existing between attachment and resilience in patients affected by MDD and BD.

## 2. Materials and Methods

This cross-sectional, noninterventional study was conducted at the Psychiatry Unit for Mood Disorders of Policlinico Universitario A. Gemelli between March 2018 and April 2020. The 106 study participants were either male or female, of Italian nationality, and aged between 18 and 75 years. They were divided into three groups: a first group consisting of 51 patients diagnosed with MDD according to the DSM-5-TR [25]; a second group consisting of 55 patients with DSM-5-TR [33] BD (40% type I and 60% type II); a healthy control (HC) group of 60 participants, recruited from a nonhospital population, with negative psychiatric history and negative family history for mood disorders. Clinical assessment was performed during the first visit, and the following psychometric scales were administered: the 21-item Hamilton Depression Rating Scale (HAM-D-21); the Hamilton Anxiety Rating Scale (HAM-A); the Young Mania Rating Scale (YMRS); the Snaith–Hamilton Pleasure Scale (SHAPS); the Barratt Impulsiveness Scale Version 11 (BIS-11); the 20-item Toronto Alexithymia Scale (TAS-20); the Connor–Davidson Resilience Scale (CD-RISC); and Experiences in Close Relationship (ECR), which investigates attachment styles. The study received approval from the local ethical committee (Ethics Committee of the Fondazione Policlinico Universitario Agostino Gemelli IRCCS, Università Cattolica del Sacro Cuore of Rome, Rome, Italy, ID 3275). Written informed consent was obtained from all participants.

### 2.1. Psychometric Scales

#### 2.1.1. 21-Item Hamilton Depression Rating Scale (HAM-D-21)

The Hamilton Depression Rating Scale (HAM-D-21) is the best known and most used clinician-rated scale for depression, considered the reference for research, although it is not a specific diagnostic tool for depression. It is a dimensional scale suitable for quantitatively evaluating the severity of depressive symptoms. It is indicated for adult patients with depressive symptoms of any type. The original version, consisting of 17 items, was subsequently extended to 21- and 24-item versions; however, the score count is limited to the first 17 items. The items that count are 1. Depressed mood (rated 0–4), 2. Guilt feelings (0–4), 3. Suicide (0–4), 4. Insomnia—Early (0–2), 5. Insomnia—Middle (0–2), 6. Insomnia—Late (0–2), 7. Work and interests (0–4), 8. Retardation–psychomotor (0–4), 9. Agitation (0–2), 10. Anxiety—Psychic (0–4), 11. Anxiety—Somatic (0–4), 12. Somatic symptoms—Gastrointestinal (0–2), 13. Somatic symptoms—General (0–2), 14. Sexual dysfunction/Menstrual disturbance (0–2), 15. Hypochondriasis (0–4), 16. Weight loss by history (0–2), and 17. Insight (0–2). The other items are 18. Diurnal variation (0–2), 19. Depersonalization and derealization (0–4), 20. Paranoid symptoms (0–4), and 21. Obsessional symptoms (0–2), and these do not count, despite being rated. The minimum score is 0 and the maximum is 52. The scale is sensitive to change. The total score is an expression of the pervasiveness of depression given the heterogeneity of the items, and it is calculated by adding up the points of the first 17 items. Scores ≥25 indicate severe depression, 18–24 moderate, 8–17 mild, and ≤7 indicate an absence of depression [34].

#### 2.1.2. Hamilton Anxiety Rating Scale (HAM-A)

The Hamilton Anxiety Rating Scale (HAM-A) is a clinician-report tool originally intended to assess individuals “already diagnosed with anxiety neurosis” [35]. It is composed of 14 items, each of which actually represents a cluster in which different symptoms are grouped, from two to eight, associated with each other either by their nature or because clinical experience indicates that they are related to each other. Factor analysis extracted two factors: Somatic Anxiety, which includes items 7 to 13 (7. Somatic (muscular), 8. Somatic (sensory), 9. Cardiovascular symptoms, 10. Respiratory symptoms, 11. Gastrointestinal symptoms, 12. Genitourinary symptoms, and 13. Autonomic symptoms), and Psychic Anxiety, consisting of the first 6 items and the 14th item (1. Anxious mood, 2. Tension, 3. Fears, 4. Insomnia, 5. Intellectual, 6. Depressed mood, and 14. Behavior at interview). Each item is rated on a 5-point Likert scale, where 0 is “absent”, 1 “mild”, 2 “moderate”, 3 “severe”, and 4 “very severe”. The total score on the scale can therefore range from 0 to 56, where <17 indicates mild, 18–24 mild to moderate, and 25–30 moderate to severe anxiety [35].

#### 2.1.3. Young Mania Rating Scale (YMRS)

The Young Mania Rating Scale (YMRS) is a clinician-rated scale which is suitable for the evaluation of adult patients with manic symptoms of varying severity. It should only be used as a quantitative assessment tool for mania and not as a diagnostic tool. It is composed of 11 items that accurately explore key manic symptoms. The assessment is based both on what the patient reports about his/her condition in the previous 48 h and on what the clinician observes of the patient’s behavior during the interview, with the latter being more important than the former. The items include 1. Elevated Mood, 2. Increased Motor Activity/Energy, 3. Sexual Interest, 4. Sleep, 7. Language–Thought Disorder, 10. Appearance, and 11. Insight, which are rated on a 5-point Likert scale with scores of 0, 1, 2, 3, and 4. The items 5. Irritability, 6. Speech (Rate and Amount), 8. Content, and 9. Disruptive–Aggressive Behavior are rated 0, 2, 4, 6, and 8, according to increasing severity. The scores range from 0 to 60. A total score ≥13 represents a potential case of mania or hypomania, while ≥21 indicates a probable case of mania or hypomania [36].

#### 2.1.4. Snaith–Hamilton Pleasure Scale (SHAPS)

The Snaith–Hamilton Pleasure Scale (SHAPS) is a self-assessment scale composed of 14 items that explore interests, social interactions, and sensory experiences (1. I would enjoy my favourite television or radio programme; 2. I would enjoy being with my family or close friends; 3. I would find pleasure in my hobbies and pastimes; 4. I would be able to enjoy my favourite meal; 5. I would enjoy a warm bath or refreshing shower; 6. I would find pleasure in the scent of flowers or the smell of a fresh sea breeze or freshly baked bread; 7. I would enjoy seeing other people’s smiling faces; 8. I would enjoy looking smart when I have made an effort with my appearance; 9. I would enjoy reading a book, magazine or newspaper; 10. I would enjoy a cup of tea or coffee or my favourite drink; 11. I would find pleasure in small things, e.g., bright sunny day, a telephone call from a friend; 12. I would be able to enjoy a beautiful landscape or view; 13. I would get pleasure from helping others; and 14. I would feel pleasure when I receive praise from other people). The participant has to declare whether he/she agrees or disagrees (totally or fairly) with what is expressed in each item. The total score on this scale can range from 0 to 14, and scores ≥ 3 indicate a significant reduction in hedonic abilities [37].

#### 2.1.5. Barratt Impulsiveness Scale Version 11 (BIS-11)

The Barratt Impulsiveness Scale Version 11 (BIS-11) is one of the most used tools for the study of impulsiveness; it is a 30-item self-assessment questionnaire, with each item rated on a 4-point Likert scale ranging from 1 (“never/rarely”) to 4 (“almost always/always”). The items are questions which the completer poses to oneself (1. I plan tasks carefully? 2. I do things without thinking? 3. I make-up my mind quickly? 4. I am happy-go-lucky? 5. I don’t “pay attention”? 6. I have “racing” thoughts? 7. I plan trips well ahead of time? 8. I am self-controlled? 9. I concentrate easily? 10. I save regularly? 11. I “squirm” at plays or lectures? 12. I am a careful thinker? 13. I plan for job security? 14. I say things without thinking? 15. I like to think about complex problems? 16. I change jobs? 17. I act “on impulse”? 18. I get easily bored when solving thought problems? 19. I act on the spur of the moment? 20. I am a steady thinker? 21. I change residences? 22. I buy things on impulse? 23. I can only think about one thing at a time? 24. I change hobbies? 25. I spend or charge more than I earn? 26. I often have extraneous thoughts when thinking? 27. I am interested in the present than the future? 28. I am restless at the theater or lectures? 29. I like puzzles? 30. I am future oriented?). The total score ranges from 30 to 120 and offers a quantitative estimate of impulsivity that derives from the sum of three factors, i.e., cognitive impulsivity (minimum score: 8; maximum: 32), motor impulsivity (minimum 11; maximum 44) and non-planning impulsivity (minimum 11; maximum 44) [38]. The higher the score, the greater the impulsivity.

#### 2.1.6. Toronto Alexithymia Scale (TAS-20)

The Toronto Alexithymia Scale (TAS-20) is a self-report questionnaire consisting of 20 items, each rated on a 5-point Likert scale from 1 to 5 (strongly disagree to strongly agree). It measures the three dimensions that define the construct of alexithymia, i.e., difficulty in identifying feelings and distinguishing between feelings and physical sensations (DIF). Alexithymia is identified by the following items: 1. I am often confused about what emotion I am feeling; 3. I have physical sensations that even doctors don’t understand, 6. When I am upset, I don’t know if I am sad, frightened, or angry; 7. I am often puzzled by sensations in my body; 9. I have feelings that I can’t quite identify; 13. I don’t know what’s going on inside me; and 14. I often don’t know why I am angry. The difficulty in describing one’s feelings to others (DDF) is identified by the following items: 2. It is difficult for me to find the right words for my feelings; 4. I am able to describe my feelings easily; 11. I find it hard to describe how I feel about people; 12. People tell me to describe my feelings more; and 17. It is difficult for me to reveal my innermost feelings, even to close friends. The cognitive style oriented toward external reality (EOT) is identified by the following items: 5. I prefer to analyze problems rather than just describe them; 8. I prefer to just let things happen rather than to understand why they turned out that way; 10. Being in touch with emotions is essential; 15. I prefer talking to people about their daily activities rather than their feelings; 16. I prefer to watch “light” entertainment shows rather than psychological dramas; 18. I can feel close to someone, even in moments of silence; 19. I find examination of my feelings useful in solving personal problems; and 20. I look for hidden meanings in movies or plays. The TAS-20 showed adequate internal reliability; test–retest; and factorial, convergent, and discriminant validity. Possible total scores range 20 to 100, with <51 meaning no alexithymia, 51-60 meaning borderline alexithymia, and a total score ≥ 61 indicating alexithymia [39,40].

#### 2.1.7. Connor–Davidson Resilience Scale (CD-RISC)

The Connor–Davidson Resilience Scale (CD-RISC) is a self-assessment scale that was created with the aim of using it in a valid and reliable way to detect resilience. It can also be used to detect changes in resilience levels due to pharmacological or psychotherapeutic treatments. The CD-RISC consists of 25 items and 5 factors, i.e., personal competence, high standards, tenacity (items include 10. Best effort no matter what; 11. You can achieve your goals; 12. When things look hopeless, I don’t give up; 16. Not easily discouraged by failure; 17. Think of self as strong person; 23. I like challenges; 24. You work to attain your goals; and 25. Pride in your achievements), trust in one’s instincts, tolerance of negative affect, and strengthening effects of stress (items include 6. See the humorous side of things; 7. Coping with stress strengthens; 14. Under pressure, focus and think clearly; 15. Prefer to take the lead in problem solving; 18. Make unpopular or difficult decisions; 19. Can handle unpleasant feelings; and 20. Have to act on a hunch), positive acceptance of change and secure relationships (items include 1. Able to adapt to change; 2. Close and secure relationships; 4. Can deal with whatever comes; 5. Past success gives confidence for new challenge; and 8. Tend to bounce back after illness or hardship), control (items include 13. Know where to turn for help; 21. Strong sense of purpose; and 22. In control of your life), and spiritual influences (items include 3. Sometimes fate or God can help and 9. Things happen for a reason). It is based on a 5-point Likert scale, ranging from 0 (“Not true at all”) to 4 (“True nearly all the time”), with possible scores ranging between 0 and 100. Total scores range from 0 to 100; the higher the score, the higher the resilience level [41]. The authors do not recommend calculating individual factor scores [42].

#### 2.1.8. Experiences in Close Relationship (ECR)

Experiences in Close Relationship (ECR) is a self-assessment scale used in measuring adult romantic attachment. It consists of 36 items, measured on a 7-point Likert scale ranging from 1 (“Strongly disagree”) to 7 (“Strongly agree”), and is composed of two independent dimensions, i.e., attachment-related anxiety (items include 1. I’m afraid that I will lose my partner’s love; 2. I often worry that my partner will not want to stay with me; 3. I often worry that my partner doesn’t really love me; 4. I worry that romantic partners won’t care about me as much as I care about them; 5. I often wish that my partner’s feelings for me were as strong as my feelings for him or her; 6. I worry a lot about my relationships; 7. When my partner is out of sight, I worry that he or she might become interested in someone else; 8. When I show my feelings for romantic partners, I’m afraid they will not feel the same about me; 9. I rarely worry about my partner leaving me; 10. My romantic partner makes me doubt myself; 11. I do not often worry about being abandoned; 12. I find that my partner(s) don’t want to get as close as I would like; 13. Sometimes romantic partners change their feelings about me for no apparent reason; 14. My desire to be very close sometimes scares people away; 15. I’m afraid that once a romantic partner gets to know me, he or she won’t like who I really am; 16. It makes me mad that I don’t get the affection and support I need from my partner; 17. I worry that I won’t measure up to other people; and 18. My partner only seems to notice me when I’m angry) and attachment-related avoidance (items include 19. I prefer not to show a partner how I feel deep down; 20. I feel comfortable sharing my private thoughts and feelings; 21. I find it difficult to allow myself to depend on romantic partners; 22. I am very comfortable being close to romantic partners; 23. I don’t feel comfortable opening up to romantic partners; 24. I prefer not to be too close to romantic partners; 25. I get uncomfortable when a romantic partner wants to be very close; 26. I find it relatively easy to get close to my partner; 27. It’s not difficult for me to get close to my partner; 28. I usually discuss my problems and concerns with my partner; 29. It helps to turn to my romantic partner in times of need; 30. I tell my partner just about everything; 31. I talk things over with my partner; 32. I am nervous when partners get too close to me; 33. I feel comfortable depending on romantic partners; 34. I find it easy to depend on romantic partners; 35. It’s easy for me to be affectionate with my partner; and 36. My partner really understands me and my needs). Scores on items 9 and 11 were reversed, i.e., 1 counted as 7, 2 counted as 6, 3 was 5, 5 became 3, 6 was 2, and 7 was rated 1, while 4 remained 4. We randomized the order of items when administering the questionnaires to conform with developers’ recommendations for performing research. Total scores range from 36 to 252, 18 to 126 for each dimension. Combined, the two scales yield Bartholomew’s attachment styles [43], i.e., secure (positive models of self and others), anxious–preoccupied (negative model of self and positive model of others), dismissive–avoidant (positive model of self and negative model of others), and fearful–avoidant (negative models of self and others) [44].

### 2.2. Statistical Analyses

Statistical analyses were performed using IBM-SPSS statistics software, version 24.0 (IBM Corporation, Armonk, NY, USA, 2016), by an independent researcher who did not know the study participants. The categorical variables are given as numbers and percentages; the continuous variables are given as mean ± standard deviation (SD). The ANOVA test was used to conduct a preliminary analysis among the three groups. Pearson’s correlation analysis was used to compare continuous variables, the chi-square test to compare categorical variables, and post-hoc analyses for group-to-group comparisons. Statistical significance was set at *p* < 0.05.

## 3. Results

The clinical and socio-demographic characteristics of the two clinical samples and HCs are shown in Table 1. Significant differences were found between the mood disorders and the HC samples in schooling (*p* = 0.001), with HCs having higher educational levels, while the two clinical samples differed little.

The ages of the male and female participants did not differ from one another (N = 64; mean = 51 ± 11.9 years vs. N = 102, mean 52 ± 10.85 years, respectively; *t* = −0.55, *p* = 0.58, not significant, n.s.). Male and female scores on the HAM-D-21 (mean 8.27 ± 7.01 vs. 8.75 ± 7.26, respectively; *t* = 0.49; *p* = 0.7), HAM-A (mean 7.20 ± 7.68 vs. 8.48 ± 7.54, respectively; *t* = 1.05; *p* = 0.3), YMRS (mean 2.27 ± 2.95 vs. 2.22 ± 2.08, respectively; *t* = −0.13; *p* = 0.9), BIS-11 (mean 62.03 ± 10.47 vs. 62.97 ± 9.89, respectively; *t* = −0.58; *p* = 0.56), TAS-20 (mean 49.14 ± 13.67 vs. 49.36 ± 13.26, respectively; *t* = −0.1; *p* = 0.9), CD-RISC (mean 57.97 ± 17.47 vs. 57.75 ± 15.51, respectively; *t* = 0.09; *p* = 0.93), ECR Attachment-related avoidance (mean 46.87 ± 21.13 vs. 50.40 ± 21.65, respectively; *t* = −1.03; *p* = 0.30), and ECR Attachment-related anxiety (mean 63.64 ± 23.55 vs. 68.75 ± 22.25, respectively; *t* = −1.40; *p* = 0.16) did not differ, but males scored higher on the SHAPS than females (mean 1.89 ± 2.59 vs. 1.17 ± 1.85, respectively; *t* = 2.1; *p* = 0.04).

Table 2 shows the results obtained for patients with MDD, BD, and HCs in tests aimed at investigating mood-related symptoms (HAM-D-21, HAM-A, YMRS, SHAPS, BIS-11, and TAS-20).

Patients with MDD and BD scored on the HAM-D-21 in a range indicating mild depression in both cases, while HCs scored low and their scores were compatible with an absence of depression. ANOVA showed that patients with mood disorders scored significantly higher than HCs on the HAM-D-21 (*p* < 0.0001). Similar scores were obtained for the HAM-A, with the scores of the clinical groups indicating mild anxiety.

All groups scored low on the YMRS, with scores indicating the absence of mania and hypomania, but patients with BD scored higher than MDD, while HCs scored lower than both clinical groups (Table 2).

On the total SHAPS test, HCs scored lower than both clinical groups; BD scored higher than HCs on all SHAPS dimensions, while MDD scored higher than HCs only on the interest and sensory experiences subscales. However, MDD and BD did not differ significantly on any subscale.

Both patient groups scored higher on total impulsivity than HCs, with no difference between them. Patients with MDD did not differ from patients with BD on any of the BIS-11 subscales, and all differed from HCs on the BIS-11 dimensions, but differently. MDD scored higher than HCs on the attentional impulsivity subscale, while BD scored higher than HCs on both motor and non-planning impulsivity (Table 2).

Both clinical groups scored higher than HCs on total alexithymia, but the results differed for their scores on the TAS-20 subscales. The MDD group scored higher than either the BD or the HC group in the dimensions of difficulty in identifying feelings and difficulty in communicating feelings to others, while they scored higher than HCs only in the outward thinking dimension. Patients with MDD scored higher than patients with BD in the dimensions of difficulty in identifying feelings and difficulty in communicating feelings to others, while patients with BD and MDD scored higher than HCs in the outward-oriented thinking dimension. In this dimension, the two clinical groups did not differ, but the MDD group scored higher than the BD group on the total TAS-20 scale.

The results relative to resilience and attachment are shown in Table 3. The two patient groups did not differ in their scores for total resilience or for any of the CD-RISC factors. HCs scored higher (indicating better resilience) than patients with MDD for total CD-RISC and for all subscales apart from control, whereas they scored better than BD patients for total CD-RISC and for the factors of personal competence, high standards, tenacity, trust in one’s instincts, tolerance of negative affect, strengthening effects of stress, positive acceptance of change, and secure relationships. However, since the authors do not recommend calculating separate scores for the factors, these results are of limited value.

Both patient samples scored significantly higher on the attachment-related anxiety and attachment-related avoidance factors of the ECR than HCs (*p* < 0.0001 in all cases). There were no significant differences between the MDD and the BD group regarding the ECR scales. While the MDD and BD groups did not differ in their Bartholomew attachment styles (*χ*^2^ = 5.38; *p* = 0.146), both groups differed from the HCs (*χ*^2^ = 78.62; *p* < 0.00001) in that most people in the latter group were shown to have a secure attachment style.

Of the 55 patients with BD, 22 had BD-I (10 males and 12 females) and 33 had BD-II (15 males and 18 females). The ages of the BD-I patients did not differ from the BD-II group (N = 22; mean = 51.5 ± 13.08 years vs. N = 33, mean 51.2 ± 9.86 years, respectively; *t* = 0.09, *p* = 0.93, n.s.). The scores of the BD-I patients on the HAM-D-21 (mean 8.73 ± 6.21 vs. 10.76 ± 7.88, respectively; *t* = −1.02; *p* = 0.31), HAM-A (mean 8.64 ± 6.79 vs. 9.45 ± 8.95, respectively; *t* = −0.36; *p* = 0.7), YMRS (mean 4.27 ± 3.28 vs. 3.27 ± 2.54, respectively; *t* = 1.27; *p* = 0.21), SHAPS (mean 2.18 ± 2.17 vs. 2.39 ± 2.9, respectively; *t* = −0.29; *p* = 0.77), BIS-11 (mean 63.86 ± 9.46 vs. 65.09 ± 11.25, respectively; *t* = −0.42; *p* = 0.68), TAS-20 (mean 47.32 ± 10.42 vs. 52.48 ± 14.29, respectively; *t* = −1.45; *p* = 0.15), CD-RISC (mean 59.18 ± 14.14 vs. 51.91 ± 15.41, respectively; *t* = 1.77; *p* = 0.08), ECR attachment-related avoidance (mean 56.27 ± 26.49 vs. 58.27 ± 19.42, respectively; *t* = −0.32; *p* = 0.75), and ECR attachment-related anxiety (mean 76.5 ± 24.04 vs. 74.91 ± 24.66, respectively; *t* = 0.24; *p* = 0.81) did not differ from those of the patients with BD-II. Among the 51 patients with MDD, there were 6 patients with a positive family history for MDD, 3 with a positive family history for BD, and 1 with a positive family history for schizophrenia; among the 22 patients with BD-I, family history was positive for MDD in 2 patients, for BD in 5 patients, and for schizophrenia in 2 patients. while among the 33 patients with BD-II, family history was positive for MDD in 2 patients, for BD in 4 patients, and for schizophrenia in 2 patients. There were no statistical differences among MDD, BD-I, and BD-II patients in positivity for family history (χ^2^ = 0.97, *p* = 0.61).

Regarding Pearson’s correlations, in patients with MDD, the avoidance scores on the ECR did not significantly correlate with the CD-RISC scores. The attachment-related anxiety ECR scores negatively correlated with those for CD-RISC positive acceptance of change and secure relationships (*r* = −0.355; *p* < 0.05). In MDD, the different Bartholomew typologies did not differ from each other with regard to the CD-RISC. In patients with BD, CD-RISC personal competence, high standards, and tenacity scores correlated negatively with attachment-related avoidance on the ECR (*r* = −0.286; *p* < 0.05) as well as with attachment-related anxiety on the ECR (*r* = −0.375; *p* < 0.01); the latter correlated negatively with CD-RISC positive acceptance of change and secure relationships (*r* = −0.367; *p* <0.01), CD_RISC control (*r* = −0.367; *p* < 0.01) and total CD-RISC scores (*r* = −0.368; *p* < 0.01). In BD patients, Bartholomew’s different attachment styles differed significantly from each other in terms of CD-RISC personal competence, high standards, and tenacity (*p* = 0.016); CD-RISC control (*p* = 0.015); and total CD-RISC score (*p* = 0.038). In particular, patients with secure attachment styles had significantly higher scores for CD-RISC personal competence, high standards, and tenacity (*p* = 0.043) compared to patients with fearful–avoidant attachment styles.

## 4. Discussion

The present study revealed significant differences between patients with mood disorders and non-clinical controls, both in terms of resilience and attachment styles. Patients with MDD and BD (types I and II) displayed significantly lower resilience than HCs. The results confirm the data found in the literature [14,45,46]. Patients with MDD and BD showed secure attachment in 27.4% and 18.2% of cases, respectively, while HCs showed it in 90% of cases. In both groups of patients with mood disorders, fearful attachment prevailed; it was present in more than half (60%) of patients with BD and in 39.2% of those with MDD. Our data concerning attachment styles seem to confirm previous data showing that patients with mood disorders have significantly higher rates of insecure attachment than HCs [28,47,48,49,50].

In this study, we found that patients with MDD and BD differed little on scales focusing on mood symptoms, impulsivity, the ability to perceive pleasure, and the ability to find words for their emotions. In particular, both patient groups scored in the mild range of depression, anxiety, and mania/hypomania, which means that they were evaluated during euthymic phases. All patient groups scored significantly higher on all scales than HCs, but this was an expected outcome. Gender did not appear to affect results, save for a higher tendency of males to show anhedonia than females. Comparing BD-I patients with the BD-II group, we found no differences to occur on any of the scales. It is possible that the subdivision of the BD sample could have resulted in samples too small for differences to be detected, and this also holds true for family history.

While it is easy to hypothesize that secure attachment, through the introjection of good objects, is able to increase resilience [51,52], the biological underpinnings are unclear at the moment, although they are the object of investigation. Resilience was found to relate to effective hypothalamic–pituitary–adrenal (HPA) axis function in one early study, with more resilient subjects being able to restrict cortisol secretion [53] and to rely on proper functioning of integrated parallel emotion-regulating circuits involving the amygdala, the hippocampus, the anterior cingulate cortex, and the ventro-medial prefrontal cortex [54]. Similarly, attachment was found to relate to appropriate cortisol secretion with exposure to stress [54], and was reported to be dysfunctional (insecure) when the amygdala was hypofunctional [55,56]. The function of a neural circuit comprising the amygdala, the inferior frontal gyrus, the anterior cingulated cortex, and the hippocampus has been related to attachment [57], thus pointing to functional commonalities with resilience. Furthermore, oxytocin has been proposed as and found to be the attachment and affiliation hormone/neuromodulator [58,59], and the same peptide has been shown to be involved in resilience [60,61,62]. Taken together, all these considerations provide theoretical support to our findings and point to resilience and attachment as two strictly interconnected systems.

Although associations have been found between resilience and attachment, and between the latter and depression [22,28], there are no studies aiming to investigating the relationship between resilience and attachment in mood disorders. In the present study, resilience significantly correlated with attachment, particularly in patients with BD. In these patients, both dimensions of the ECR, i.e., avoidance and anxiety, correlated significantly and negatively with resilience. On the other hand, in patients with MDD, the only dimension of attachment-related anxiety correlated significantly and negatively with resilience, and, in particular, with the positive acceptance of change dimensions. These results confirm the central role played by early life experiences and attachment relationships in patients with mood disorders. Consequently, an adequate treatment for mood disorders should not only focus on the intrapsychic and cognitive aspects of the psychiatric problem, but should also address the analysis and correction of the patient’s interpersonal relational models. This is supported by the efficacy of interpersonal psychotherapy (IPT) in the treatment of mood disorders [63,64]. The correction of dysfunctional relational models, in fact, could represent an effective intervention modality to improve resilience and, consequently, the symptomatology and quality of life of patients.

### 4.1. Limitations

Our study did not allow us to speculate on the causal relationships between attachment and resilience due to its cross-sectional design. Furthermore, the three comparative groups were not of a sample size sufficient to obtain statistically strong results. However, the statistical significance of the obtained results and support from literature leave no doubt as to the lower resilience and lower secure attachment rates of patients with mood disorders. Other limitations include pooling BD subtypes in analyses and not including trauma- or stress-related scales in our assessment. Hence, our data should be interpreted cautiously.

### 4.2. Conclusions and Future Perspectives

The results of this study confirm those of the existing literature focusing on resilience and attachment styles in patients with MDD and BD, suggesting that people with mood disorders have lower levels of resilience and a significantly higher rate of insecure attachment than the general population. The obtained results also seem to support data deriving from previous studies which show a significant positive correlation between the quality of attachment and the development of resilience, strengthening the hypothesis that attachment constitutes the nuclear factor of resilience.

These results require confirmation in a larger sample of patients. Further studies are needed in order to evaluate the effect, over time, of the analysis and correction of dysfunctional relational models on resilience and, consequently, on quality of life. Future studies should prospectively explore resilience in patients with depressive and bipolar disorders who are undergoing treatment with psychotherapeutic techniques focusing on inward working models and addressing pathogenic beliefs. This will help with individualizing treatment strategies in mood disorder patients according to their attachment styles; such strategies should aim to strengthen resilience.

## Figures and Tables

**Table 1 jpm-13-00969-t001:** Clinical and socio-demographic characteristics of the entire sample (N = 166).

Variable	MDD (N = 51)	BD (N =55)	HCs (N = 60)	*p* Value
**Gender, M/F, N (%)**	M = 15 (29%) F = 36 (71%)	M = 25 (45%) F = 30 (54%)	M = 24 (40%) F = 36 (60%)	=0.228
**Age, years, mean ± SD**	52.7 ± 12.1	51.3 ± 11.1	52.1 ± 10.7	=0.819
**Marital status (%)**	Unmarried maiden = 7 (14%) Married/cohabiting = 34 (67%) Separated/divorced = 7 (14%) Widower = 3 (6%)	N = 13 (24%) N = 30 (55%) N = 10 (18%) N = 2 (4%)	N = 11 (18%) N = 40 (67%) N = 7 (12%) N = 2 (3%)	=0.728
**Living status (%)**	Alone = 8 (16%) Family of origin = 4 (7.8%) Own family = 39 (76.5%) Other = 0 (0%)	N = 15 (27%) N = 5 (9%) N = 33 (60%) N = 2 (4%)	N = 10 (17%) N = 4 (7%) N = 43 (72%) N = 3 (5%)	=0.423
**Education (%)**	Primary school = 1 (2%) Middle school = 12 (23.5%) High school = 23 (45.1%) University = 15 (29.4%)	N = 0 (0%) N = 14 (25%) N = 32 (58%) N = 9 (16%)	N = 1 (2%) N = 4 (7%) N = 20 (33%) N = 35 (58%)	**=0.001**
**Profession (%)**	Student = 2 (3.9%) Employee = 19 (37.3%) Freelance = 14 (27.4%) Unemployed = 16 (31.4%)	N = 2 (3.6%) N = 20 (36.4%) N = 15 (27.3%) N = 18 (32.7%)	N = 2 (3.3%) N = 25 (41.7%) N = 22 (36.7%) N = 11 (18.3%)	=0.672
**Only child (%)**	Yes = 20 (39%) No = 31 (61%)	N = 17 (30.9%) N = 38 (69.1%)	N = 13 (21.7%) N = 47 (78.3%)	=0.131
**Previous contacts for mood disorders (%)**	Yes = 14 (27%) No = 37 (72%)	N = 24 (44%) N = 31 (56%)	N = 0 (0%) N = 60 (100%)	**<0.0001**
**Months from symptom onset to treatment, mean ± SD**	63.4 ± 66.9	107 ± 119.8	-	**<0.0001**
**Ongoing psychopharmacotherapy**	Yes = 3 (5.9%) No = 48 (94.1%)	N = 4 (7.3%) N = 51 (92.7%)	N = 0 (0%) N = 60 (100%)	**<0.0001**
**Ongoing psychotherapy**	Yes = 8 (15.7%) No = 43 (84.3%)	N = 7 (12.7%) N = 48 (87.3%)	N = 0 (0%) N = 60 (100%)	**<0.0001**
**Previous psychotherapy**	Yes = 6 (11.8%) No = 45 (88.2%)	N = 9 (16.4%) N = 46 (83.6%)	N = 0 (0%) N = 60 (100%)	**<0.0001**
**Months of current/past psychotherapy, mean ± SD**	6.1 ± 19.8	6.3 ± 16.6	-	**<0.0001**

All significant results in **bold**. Abbreviations: BD, bipolar disorder; F, female; HCs, healthy controls; M, male; MDD, major depressive disorder; SD, standard deviation.

**Table 2 jpm-13-00969-t002:** Scores of the three samples on the mood-centered psychometric scales (total sample, N = 166).

	MDD (N = 51)	BD (N = 55)	HCs (N = 60)	*p* Value
**HAM-D** **-21** **total, mean** **±** **SD**	13.8 ± 6.2	9.9 ± 7.3	2.9 ± 1.7	MDD vs. BD: =1.000; MDD vs. HCs: <**0.0001**; BD vs. HCs: <**0.0001**
**HAM-A PSYC total, mean ± SD**	8.1 ± 5.1	6.4 ± 4.7	2.5 ± 1.8	MDD vs. BD: 0.095; MDD vs. HCs: <**0.0001**; BD vs. HCs: <**0.0001**
**HAM-A SOM total, mean ± SD**	3.9 ± 4.2	2.7 ± 3.8	1 ± 1.1	MDD vs. BD: 0.176; MDD vs. HCs: <**0.0001**; BD vs. HCs: =**0.019**
**HAM-A total, mean ± SD**	12 ± 8.5	9.1 ± 8.1	3.5 ± 2.1	MDD vs. BD: 0.083; MDD vs. HCs: <**0.0001**; BD vs. HCs: <**0.0001**
**YMRS total, mean ± SD**	2.5 ± 2	3.7 ± 2.9	0.7 ± 1.1	MDD vs. BD: **0.017**; MDD vs. HCs: **0.000**; BD vs. HCs: <**0.0001**
**SHAPS interests, mean ± SD**	0.7 ± 0.9	0.9 ± 1	0.3 ± 0.4	MDD vs. BD: =0.367; MDD vs. HCs: =**0.032**; BD vs. HCs: <**0.0001**
**SHAPS eat and drink, mean ± SD**	0.1 ± 0.3	0.2 ± 0.5	0.1 ± 0.2	MDD vs. BD: =0.297; MDD vs. HCs: =1.000; BD vs. HCs: =**0.022**
**SHAPS social interactions, mean ± SD**	0.3 ± 0.7	0.5 ± 0.8	0.1 ± 0.4	MDD vs. BD: =0.839; MDD vs. HCs: =0.261; BD vs. HCS: =**0.013**
**SHAPS sensory experiences, mean ± SD**	0.6 ± 1.1	0.7 ± 1.2	0.1 ± 0.2	MDD vs. BD: =1.000; MDD vs. HCs: =**0.010**; BD vs. HCs: =**0.001**
**SHAPS total, mean ± SD**	1.7 ± 2.4	2.3 ± 2.6	0.5 ± 0.7	MDD vs. BD: =0.327; MDD vs. HCs: =**0.007**; BD vs. HCs: <**0.0001**
**BIS-11, attentional impulsivity, mean ± SD**	17.3 ± 4.6	15.5 ± 4	15.2 ± 3.6	MDD vs. BD: =0.064; MDD vs. HCs: =**0.021**; BD vs. HCs: =1.000
**BIS-11, motor impulsivity, mean ± SD**	20.6 ± 4.5	21.4 ± 4.7	19.3 ± 3.9	MDD vs. BD: =1.000; MDD vs. HCs: =0.373; BD vs. HCs: =**0.034**
**BIS-11, non-planning impulsivity, mean ± SD**	26.6 ± 5	27.7 ± 5	24.6 ± 4.4	MDD vs. BD: =0.770; MDD vs. HCs: =0.085; BD vs. HCs: =**0.002**
**BIS-11 total, mean ± SD**	64.6 ± 10.6	64.6 ± 10.5	59.1 ± 8.3	MDD vs. BD: =1.000; MDD vs. HCs: =**0.013**; BD vs. HCs: =**0.010**
**TAS-20,** **difficulty in identifying feelings, mean** **± SD**	15.7 ± 4.1	12.7 ± 4.9	12.7 ± 3.9	MDD vs. BD: =**0.001**; MDD vs. HCs: =**0.001**; BD vs. HCs: =1.000
**TAS-20,** **difficulty in communicating feelings to others, mean** **± SD**	21.6 ± 7.4	17.4 ± 7.5	13.9 ± 5.9	MDD vs. BD: =**0.006**; MDD vs. HCs: <**0.0001**; BD vs. HCs: =**0.024**
**TAS-20, outward-oriented thinking, mean ± SD**	19.1 ± 6.3	20.4 ± 4.5	15.5 ± 4.9	MDD vs. BD: =0.609; MDD vs. HCs: =**0.002**; BD vs. HCs: <**0.0001**
**TAS-20 total, mean ± SD**	56.4 ± 13.6	50.4 ± 13	42.2 ± 9.6	MDD vs. BD: =**0.035**; MDD vs. HCs: <**0.0001**; BD vs. HCs: =**0.001**

All significant results are in **bold**. *Abbreviations:* BD, bipolar disorder; BIS-11, Barratt Impulsiveness Scale Version 11 (range 30–120); HAM-A, Hamilton Anxiety Rating Scale (range 0–56); HAM-D-21, 21-item Hamilton Depression Rating Scale (range 0–52); HCs, healthy controls; MDD, major depressive disorder; PSYC, psychic anxiety factor of the HAM-A (mental agitation and psychological distress); SHAPS, Snaith–Hamilton Pleasure Scale (range 0–14); SD, standard deviation; SOM, somatic anxiety (physical complaints related to anxiety); TAS-20, Toronto Alexithymia Scale (range 20–100); YMRS, Young Mania Rating Scale (range 0–60).

**Table 3 jpm-13-00969-t003:** Scores on the resilience and attachment scales (total sample, N = 166).

Scale Dimensions	MDD (N = 51)	BD (N = 55)	HCs (N = 60)	*p* Value
**CD-RISC Personal competence, high standards, and tenacity, mean ± SD**	16.2 ± 7.3	17.9 ± 6.6	21.9 ± 4.7	ANOVA MDD vs. BD: =0.447; MDD vs. HCs: <**0.0001**; BD vs. HCs: =**0.003**
**CD-RISC Trust in one’s instincts, tolerance of negative affect, and strengthening effects of stress, mean ± SD**	13.9 ± 5.5	14.9 ± 6.3	18.1 ± 4.4	MDD vs. BD: =1.000; MDD vs. HCs: <**0.0001**; BD vs. HCs: =**0.006**
**CD-RISC Positive acceptance of change and secure relationships, mean ± SD**	10.5 ± 3.8	10.7 ± 3.4	14.1 ± 3.2	MDD vs. BD: =1.000; MDD vs. HCs: <**0.0001**; BD vs. HCs: <**0.0001**
**CD-RISC Control, mean ± SD**	6.7 ± 2.6	7.2 ± 2.7	8.2 ± 2.3	MDD vs. BD: =0.795; MDD vs. HCs: =**0.005**; BD vs. HCs: =0.120
**CD-RISC Spiritual influences, mean ± SD**	4.1 ± 2.1	4.2 ± 2.2	3.9 ± 2.3	MDD vs. BD: =1.000; MDD vs. HCs: =1.000; BD vs. HCs: =1.000
**CD-RISC total, mean ± SD**	51.4 ± 17.7	54.8 ± 15.2	66.1 ± 12.1	MDD vs. BD: =0.720; MDD vs. HCs: <**0.0001**; BD vs. HCs: <**0.0001**
**ECR Attachment-related avoidance, mean ± SD**	53.6 ± 21.7	57.5 ± 22.3	37.5 ± 14.6	MDD vs. BD: =0.932; MDD vs. HCs: <**0.0001**; BD vs. HCs: <**0.0001**
**ECR Attachment-related anxiety, mean ± SD**	72.3 ± 22	72.6 ± 24.2	54 ± 15.7	MDD vs. BD: =1.000; MDD vs. HCs: <**0.0001**; BD vs. HCs: <**0.0001**
**ECR by Bartholomew type, N (%)**				*χ* ^2^
Secure	14 (27.4%)	10 (18.2%)	54 (90%)	MDD vs. BD: =0.146; MDD vs. HCs: <**0.0001**; BD vs. HCs: <**0.0001**
Fearful–avoidant	20 (39.2%)	33 (60%)	1 (1.7%)	
Dismissive–avoidant	10 (19.6%)	5 (9.1%)	4 (6.7%)	
Anxious–preoccupied	7 (13.8%)	7 (12.7%	1 (1.7%)	

All significant results are in **bold**. *Abbreviations:* ANOVA, analysis of variance; BD, bipolar disorder; CD-RISC, Connor–Davidson Resilience Scale (range 0–100); ECR, Experiences in Close Relationship (range 36–252, attachment-related avoidance 18–126, attachment-related anxiety 18–126); MDD, major depressive disorder; SD, standard deviation.

## Data Availability

Anonymized data will be provided upon reasonable request to the corresponding author.

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
