# Peer review of "Resilience and Attachment in Patients with Major Depressive Disorder and Bipolar Disorder"

_jpm, 2023, doi:10.3390/jpm13060969_

Round 1

Reviewer 1 Report

Dear authors, 

This represents an interesting study aiming to investigate the associations between attachment styles and resilience in patients with mood disorders compared to healthy controls. Although the study presents numerous behavior scales and an interesting topic, I believe that the study methodology and focus showed be narrowed. Perhaps if possible, it will be important to emphasize the limitations such as the possibility to examine thoroughly causality; thus results should be interpreted cautiously.  Additionally, I believe that to ascertain trauma a scale such as the Multidimensional Trauma Recovery & Resiliency Scale (MTRR) could be added and examining early trauma, history of bulling or any previous self-harm behavior should be important. 

With regards to the hypothesis, the introduction should elaborate more in terms of describing the network of associations between resilience and attachment theories as well as previous literature and its association with development of further disorders such as social anxiety, suicidality etc.  It will be important to summarize its relationship with mood disorders. 

Also, as of now, the study fails to summarize what has been done previously and how this study could add to the field. Critical references are missing such as Furh et al., 2017 published in Journal of Affective Disorders or Meyer et al., 2001. 

One interesting contribution can be to examine if there are any differences in gender as based in other previous literature as well as any differences in terms of Bipolar subtypes. An important addition should be to add % of family of affective disorders in each subgroup. 

Similarly, the discussion section should be further improved beyond the confirmation of previous data and explain in more detailed the psychological underpinnings and biological theories that support attachment styles and resilience. 

Dear authors, 

I will suggest proof reading from a native english speaker since there are several grammatical errors as well as sentences which structure can be improved specifically in the introduction and discussion sections.  

Author Response

Comments and Suggestions for Authors

Dear authors, 

This represents an interesting study aiming to investigate the associations between attachment styles and resilience in patients with mood disorders compared to healthy controls. Although the study presents numerous behavior scales and an interesting topic, I believe that the study methodology and focus showed be narrowed. Perhaps if possible, it will be important to emphasize the limitations such as the possibility to examine thoroughly causality; thus results should be interpreted cautiously.  Additionally, I believe that to ascertain trauma a scale such as the Multidimensional Trauma Recovery & Resiliency Scale (MTRR) could be added and examining early trauma, history of bulling or any previous self-harm behavior should be important. 

We thank you for appreciation. We added to Limitations and interpreted data cautiously. Some of your phrasing was unintelligible (“I believe that the study methodology and focus showed be narrowed”), so my response could appear to be incomplete or unfocused. If you meant that our methodological focus was narrow, we adopted this methodology for clarity, avoiding to mingle too many constructs. The Multidimensional Trauma Recovery & Resiliency Scale (MTRR) is a 135-item questionnaire reduced to 99 in its short version (3 mentions on PubMed as of May 5, 2023). The tool has not still received international acclaim, probably due to its length. We did not administer it because we don’t use it (also because our patients would kill us if we elongated the time needed to complete all questionnaires). In our department we usually use the 28-item, retrospective, self-report questionnaire, Childhood Trauma Questionnaire (CTQ) (1760 results on PubMed on the same day), but in this study we did not. We acknowledge the importance of your suggestion to explore early trauma, history of bull(y)ing or self-harm/suicidal behavior and so on, but we did not explore it.

With regards to the hypothesis, the introduction should elaborate more in terms of describing the network of associations between resilience and attachment theories as well as previous literature and its association with development of further disorders such as social anxiety, suicidality etc.  It will be important to summarize its relationship with mood disorders. 

Thank you for the suggestion. Please mind that suicidality is not a disorder. We accepted the criticism and addressed your request in Introduction.

Also, as of now, the study fails to summarize what has been done previously and how this study could add to the field. Critical references are missing such as Furh et al., 2017 published in Journal of Affective Disorders or Meyer et al., 2001. 

We're afraid we can't agree with your statement; we stated in our Introduction that the ties between depression and attachment have been exhaustively explored (3 citations) but there is dearth of studies focusing on the relationship between resilience and attachment. Furthermore, we also included impulsiveness and alexithymia in our methods, so asking us to state explicitly what our contribution adds to the field is superfluous; careful readers will soon discover what’s new and they don’t need bombastic overstatements of our study’s merit. The “critical” references you claim that we should cite are: Meyer B, Pilkonis PA, Proietti JM, Heape CL, Egan M. Attachment styles and personality disorders as predictors of symptom course. J Pers Disord. 2001;15(5):371-89. doi: 10.1521/pedi.15.5.371.19200. and “Furh et al., 2017 published in Journal of Affective Disorders”. We were unable to localize the second. We hand-searched Dr. Daniela C. Furh’s research and found her not to focus her interests on attachment. We failed to find an article by her published on the Journal of Affective Disorders. Could you please be more specific? Regarding the first paper, the Baton Rouge-based group of authors state in their Abstract that “Adult attachment styles and personality disorders (PDs) show some conceptual and empirical overlap and both may complicate the course of symptoms among psychiatric patients”. That an insecure adult attachment style may complicate one’s life is universal and holds true not only for psychiatric patients. We wouldn’t like to cite such a paper that found “Secure attachment was linked with greater relative improvement in global functioning and a more benign course of anxiety symptoms over 6 months. Borderline PD features predicted less relative improvement of depressive symptoms over 6 months”, such a self-evident evidence.

One interesting contribution can be to examine if there are any differences in gender as based in other previous literature as well as any differences in terms of Bipolar subtypes. An important addition should be to add % of family of affective disorders in each subgroup. 

We examined gender differences, there were few. We added in Results and commented in Discussion. We added data based on the BD-I/BD-II subdivision; as expected, there were relatively more nominal family occurrences of psychotic and depressive disorders in patients with BD groups than in patients with MDD, but this has not been sufficient to detect significance with the chi-square. We reported data in Results. We added in limitations that we mainly pooled BD-I and –II data in our analyses.

Similarly, the discussion section should be further improved beyond the confirmation of previous data and explain in more detailed the psychological underpinnings and biological theories that support attachment styles and resilience. 

Thank you for the suggestions. We commented according to your suggestions to the extent this was possible. We hope you will approve the revised version, where we highlighted changes using red characters.

Comments on the Quality of English Language

Dear authors, 

I will suggest proof reading from a native english speaker since there are several grammatical errors as well as sentences which structure can be improved specifically in the introduction and discussion sections.

Thank you for suggesting that we check grammar errors and sentence structure. We did this. If something was a misprint, it does not mean that all the paper was written in a flawed language. We changed some expressions, please check. If you still find language problems in our manuscript, please make some examples, so that we understand what is going wrong for you. We thank you for your suggestions that helped us to improve our manuscript.

Reviewer 2 Report

GENERAL COMMENTS

Interesting an apparently scientifically robust paper, especially with respect to the less-often-investigated aspect of “resilience”. Acceptable after minor revision.

Throughout the paper, please give values as two significant digits only, e.g., 16 plus minus 7.3 in Table 3 – or 16.2 plus minus 7.3 if you insist instead of “16.18 plus minus 7.26”.  Giving so many digits is not more precise, it is unnecessary sophistication (and, thus, bad science).

Accordingly, percentages should be given as integers, e.g., “29% instead of “29.4% in line 210, Table 1.

Please give maximal scores for each psychometric tests in the Methods section and in the tables, too.

For all tests employed, please give links to the phrasing of the individual items.

SPECIFIC ITEMS

line 31: Please state that you used the 21-item version oft the HAMD (as stated in line 114)

line 33: Please correct to “Connor-Davison Resilience Scale” (you had forgotten the “resilience”.

line 35: From WHOM did MDD and BD differ significantly? Please explicitly state that the did not differ significantly from each other.

line 36: Please explicitly state that the p value refers to CD-RISC scores.

line 39: Please change “Results” to “Our results”

line 210, Table 1: Please use integers for percentage values (e.g., “29% instead of “29.4%

The layout of Table 1 is awkward and inhibits easy reading.

line 244, Table 2: Same concerns as with respect to Table 1.

Both tables: A p value cannot be ZERO. If true, state “< 0.001” (i.e., smaller than 0.001)

line 268, Table 3: Same concerns as with respect to Table 1.

Make sure you use articles throughout and take care to spell out which parameters exactly are compared with each other and which parameters exactly are significantly different (or not) from each other.

Author Response

Review Report Form

Open Review

Quality of English Language

( ) I am not qualified to assess the quality of English in this paper
( ) English very difficult to understand/incomprehensible
( ) Extensive editing of English language required
( ) Moderate editing of English language
(x) Minor editing of English language required
( ) English language fine. No issues detected

Yes

Can be improved

Must be improved

Not applicable

Does the introduction provide sufficient background and include all relevant references?

(x)

( )

( )

( )

Are all the cited references relevant to the research?

( )

( )

( )

( )

Is the research design appropriate?

(x)

( )

( )

( )

Are the methods adequately described?

( )

(x)

( )

( )

Are the results clearly presented?

( )

(x)

( )

( )

Are the conclusions supported by the results?

(x)

( )

( )

( )

Comments and Suggestions for Authors

GENERAL COMMENTS

Interesting an apparently scientifically robust paper, especially with respect to the less-often-investigated aspect of “resilience”. Acceptable after minor revision.

Thank you for the positive consideration and attitude.

Throughout the paper, please give values as two significant digits only, e.g., 16 plus minus 7.3 in Table 3 – or 16.2 plus minus 7.3 if you insist instead of “16.18 plus minus 7.26”.  Giving so many digits is not more precise, it is unnecessary sophistication (and, thus, bad science).

We kept decimals to one in the Tables, but not regarding significance issues, it would be too awkward.

Accordingly, percentages should be given as integers, e.g., “29% instead of “29.4% in line 210, Table 1.

We provided percentages as integers.

Please give maximal scores for each psychometric tests in the Methods section and in the tables, too.

We provided possible ranges for every psychometric test. We introduced ranges in Tables’ notes.

For all tests employed, please give links to the phrasing of the individual items.

We explicitly named all items where appropriately (Methods section).

SPECIFIC ITEMS

line 31: Please state that you used the 21-item version oft the HAMD (as stated in line 114)

Yes, we used it, but it doesn’t matter, scores don’t change, it’s only the first 17 that count. We stated that the version used was the original HAM-D-21 (Hamilton, 1960) throughout the text. Please, note that despite Max Hamilton had put 21 items in his 1960 scale, he explicitly stated that only the first 17 should count.

line 33: Please correct to “Connor-Davison Resilience Scale” (you had forgotten the “resilience”.

Thank you, we corrected.

line 35: From WHOM did MDD and BD differ significantly? Please explicitly state that the did not differ significantly from each other.

Thank you for the suggestion, we complied.

line 36: Please explicitly state that the p value refers to CD-RISC scores.

We did.

line 39: Please change “Results” to “Our results”

We changed it.

line 210, Table 1: Please use integers for percentage values (e.g., “29% instead of “29.4%

The layout of Table 1 is awkward and inhibits easy reading.

We used integers for percentages throughout Table 1. We tried to improve the layout of the Tables through acting on Table margins, but the layout used by publishers’ template has severe constraints upon us.

line 244, Table 2: Same concerns as with respect to Table 1.

We explained above. We adopted similar measures for all three tables.

Both tables: A p value cannot be ZERO. If true, state “< 0.001” (i.e., smaller than 0.001)

We agree and corrected. Unfortunately, there is a trend in international publishing to disregard this.

line 268, Table 3: Same concerns as with respect to Table 1.

We responded above. We addressed your concerns for all tables. Thank you for the precious suggestions that helped us improving our manuscript. Please find changes from the preceding version in red-coloured characters.

Comments on the Quality of English Language

Make sure you use articles throughout and take care to spell out which parameters exactly are compared with each other and which parameters exactly are significantly different (or not) from each other.

We took care to use articles when needed and spelt out the measures we were comparing, as well as their statistical significance. Thank you for useful suggestions that helped us to improve our manuscript.

Submission Date                      30 March 2023

Date of this review                    05 May 2023 10:23:46

Round 2

Reviewer 1 Report

N/A

N/A

Round 3

Reviewer 1 Report

Dear authors, 

Thank you for addressing our main concerns.